# Microorganisms Involved in Hydrogen Sink in the Gastrointestinal Tract of Chickens

**DOI:** 10.3390/ijms24076674

**Published:** 2023-04-03

**Authors:** Agata Anna Cisek, Beata Dolka, Iwona Bąk, Bożena Cukrowska

**Affiliations:** 1Department of Pathology, The Children’s Memorial Health Institute, Av. Dzieci Polskich 20, 04-730 Warsaw, Poland; gutkac@op.pl; 2Department of Pathology and Veterinary Diagnostics, Institute of Veterinary Medicine, Warsaw University of Life Sciences, St. Ciszewskiego 8, 02-786 Warsaw, Poland; 3Department of Preclinical Sciences, Institute of Veterinary Medicine, Warsaw University of Life Sciences, St. Ciszewskiego 8, 02-786 Warsaw, Poland

**Keywords:** acetogens, *Campylobacter jejuni*, hydrogen uptake, methanogenic archaea, *Methanomassiliicoccus*, chicken gut microbiota, sulfate-reducing bacteria

## Abstract

Hydrogen sink is a beneficial process, which has never been properly examined in chickens. Therefore, the aim of this study was to assess the quantity and quality of microbiota involved in hydrogen uptake with the use of real-time PCR and metagenome sequencing. Analyses were carried out in 50 free-range chickens, 50 commercial broilers, and 54 experimental chickens isolated from external factors. The median values of acetogens, methanogens, sulfate-reducing bacteria (SRB), and [NiFe]-hydrogenase utilizers measured in the cecum were approx. 7.6, 0, 0, and 3.2 log_10_/gram of wet weight, respectively. For the excreta samples, these values were 5.9, 4.8, 4, and 3 log_10_/gram of wet weight, respectively. Our results showed that the acetogens were dominant over the other tested groups of hydrogen consumers. The quantities of methanogens, SRB, and the [NiFe]-hydrogenase utilizers were dependent on the overall rearing conditions, being the result of diet, environment, agrotechnical measures, and other factors combined. By sequencing of the 16S rRNA gene, archaea of the genus *Methanomassiliicoccus* (*Candidatus Methanomassiliicoccus*) were discovered in chickens for the first time. This study provides some indication that in chickens, acetogenesis may be the main metabolic pathway responsible for hydrogen sink.

## 1. Introduction

Intestinal fermentation is essential for sustaining the host’s well-being and proper functioning. It contributes to the breakdown of dietary fiber and other otherwise indigestible compounds into more accessible products, such as short-chain fatty acids (SCFAs) [1]. Hydrogen—a fermentation byproduct—when accumulated, has the ability to inhibit the regeneration of electron carriers by bacterial fermenters, and therefore adversely inhibits fermentation. The problem of hydrogen buildup appears to be solved by the process of hydrogen sink carried out by certain groups of intestinal microorganisms [2].

The production of butyrate—the most substantial energy source for colonocytes—generates large amounts of free hydrogen, larger than the formation of other SCFAs, such as propionate [1,3]. In order to ensure the continuity of this process, and to keep the proper development of intestinal villi, hydrogen sink seems essential [1]. Moreover, its role is to maintain the proper structure of the gut microbiome and homeostasis, as high concentrations of H_2_ were found to affect both the hydrogen-producing and the non-hydrogen-producing microorganisms [3]. There is also a direct link found between the dysbiosis involving the methanogenic archaea (one of the hydrogen consumers) and overall dysbiosis affecting the host’s health [4]. In a broader sense, hydrogen sink is indirectly related to the maintenance of the host’s health.

There are only a few publications mentioning the presence of hydrogen consumers in chickens, and thus the issue is poorly understood. It is known that some species of bacteria and archaea found in the intestinal microbiome of chickens can be the providers of enzymes that ensure the utilization of hydrogen, and the continuity of intestinal fermentation [5]. For instance, bacteria of the genera *Megamonas*, *Wolinella*, *Helicobacter*, and *Campylobacter* (including *C. jejuni*) produce nickel–iron [NiFe]-hydrogenases, and bacteria of the *Lachnospiraceae* family—the acetyl-coenzyme A synthase, and methanogenic archaea—produce methyl-coenzyme M reductase. Of course, the share of individual enzymes, and thus the microbionts producing them, is diverse [5]. In addition, some publications presenting this issue stand in conflict with each other: some authors argue that certain groups of microbionts—such as methanogenic archaea—do not occur in the chicken intestines, and therefore cannot affect the metabolism of hydrogen [6,7], whilst others clearly indicate their presence [8].

Only a few publications mention the existence of methanogenic archaea in the chicken gut microbiome, highlighting the dominant role of *Methanobrevibacter woesei* [9]. Experiments on rumen microbiota point out the competition between the methanogenic archaea and acetogens, and also the sulfate-reducing bacteria (SRB) [10]. No other similar phenomenon has ever been studied in chickens. On the other hand, to the best of our knowledge, only three publications mention any correlation between the bacteria of the family *Lachnospiraceae* and *C. jejuni* in chickens, but their focus was not on hydrogen sink whatsoever [11,12,13]. That is why our study addresses this issue.

The necessity of elimination of *C. jejuni* from chicken rearing is a matter of public health because campylobacteriosis is the most common cause of foodborne gastrointestinal infection in humans, and has been so since at least 2007 [14]. Broiler meat (and products thereof) is considered the main source of the human campylobacteriosis [14]. Moreover, these infections may have long-term consequences, including the Guillain–Barré syndrome [15]. In poultry, *C. jejuni* is not exactly meaningless either, since it may be correlated with the impairment of the bird’s well-being, and reduction in animal productivity [16]. After all, a lot of effort has been put into studying the competitive exclusion of *C. jejuni* by lactobacilli [1]. There is even an entire branch of industry that sells certain strains of lactic acid bacteria as probiotics. However, in birds we observe a kind of evolutional adaptation and a slight “tolerance” toward *C. jejuni*, which may indicate that perhaps this bacterium plays a certain role in the chicken gut [12]. Hypothetically, by providing enzymes for hydrogen sink, *C. jejuni* may even become useful for its host. For this reason, some part of our study has been directed at the detection and quantification of both *C. jejuni* and *Ligilactobacillus salivarius* (as a representative of the lactic acid bacteria), and their possible interactions.

In summary, the aim of this study was to identify microbionts involved in the hydrogen sink within the gastrointestinal tracts of chickens with the use of both real-time PCR and sequencing techniques. 

## 2. Results

### 2.1. The Effect of Sample Type on the Composition of Microbiota

There was a statistically significant difference between the excreta and the cecal contents in five out of six studied microbial groups (Figure 1). The median values of the methanogenic archaea, SRB, *L. salivarius*, and *C. jejuni* were higher in the excreta samples. The acetogens, on the other hand, were more abundant in the cecal content samples than in the excreta. The differences in counts of the [NiFe]-hydrogenase utilizers were negligible.

### 2.2. The Effect of Three Rearing Methods on the Composition of Microbiota

The amount of acetogens varied significantly among all three methods of chicken rearing (and the sample type). The highest, statistically significant counts of these microbes were observed in the cecal contents of the commercial chickens, followed by the cecal contents of the experimental chickens, and the excreta of the free-range chickens, which had the lowest median of acetogens (Figure 2).

When comparing the experimental vs. free-range chickens, statistically significant differences in the medians were observed in methanogenic archaea, the [NiFe]-hydrogenase utilizers, and *C. jejuni*. The abundance of these three groups was higher in the excreta of free-range chickens than in the ceca of the experimental group. Similar observations were made for the experimental vs. commercial chickens, where the counts of methanogens, [NiFe]-hydrogenase utilizers, and *C. jejuni* in the ceca of commercial chickens also exceeded those reported in the ceca of the experimental group. 

As for the SRB, statistical significance was reported between the free-range and commercial farm chickens, and for the free-range vs. experimental chickens. In both of these cases, the comparison favored the excreta of the free-range, which were the only ones that had a median value above 0.

In the case of *L. salivarius*, statistical significance was observed only in the comparison of commercial chickens to experimental and free-range chickens. The lactobacilli counts reported in the ceca of the commercial chickens were lower than those observed in the other two groups.

### 2.3. The Effect of Age on the Composition of Microbiota

In commercial chickens, acetogens were the most varying variable among the chicken age groups (Figure 3). A statistical significance was observed between week 1 and week 3, and week 1 vs. weeks 5–6. In addition, the differences between week 3 and weeks 3–4, and weeks 3–4 vs. weeks 5–6 were statistically important.

Only two statistically significant median differences were observed in the methanogenic archaea (week 1 vs. weeks 5–6, and weeks 3–4 vs. 5–6) and the [NiFe]-hydrogenase utilizers (week 3 vs. 3–4, and weeks 3–4 vs. 4–5). Counts of *C. jejuni* differed in one case, i.e., between weeks 3 and 3–4. No statistical significance among the age groups was observed in the SRB and *L. salivarius* median counts. 

By contrast, in the experimental chickens, the lactobacilli counts were the only variable that was statistically significant in relation to the chicken age (Figure 4). The median values between day 0 and days 4, 7, and 14 were significantly higher. The opposite relationship was observed between days 4 and 21. Interestingly, in this case, the younger chick had higher counts of *L. salivarius*.

As for the acetogen counts in the ceca of experimental chickens, the general trend was to decrease with chicken age, especially by the end of the experiment; however, those differences were not statistically proven. No other linear trend was observed among the age groups in either the experimental or commercial farm chickens.

The age vs. microbial populations relationship in the excreta of the free-range chickens was impossible to establish due to technical reasons. 

### 2.4. Interactions among Microbionts Involved in Hydrogen Sink

The analyses of associations among the microbionts involved in hydrogen sink showed negligible (R_s_ between 0 and 0.20) to moderate (R_s_ between 0.41 and 0.60) correlations among six studied microbial populations when the three rearing groups were tested together (Appendix A). When correlations were analyzed separately for each of the rearing groups, the highest number of statistically significant relationship results—altogether six, weakly to moderately correlated—was observed in the free-range chickens alone: methanogens vs. acetogens (R_s_ 0.38), SRB vs. *C. jejuni* (R_s_ 0.39), *L. salivarius* vs. [NiFe]-hydrogenase utilizers (R_s_ 0.57), acetogens vs. SRB (R_s_ 0.45), and SRB vs. [NiFe]-hydrogenase utilizers (R_s_ 0.41). There was only one strongly correlated relationship between the [NiFe]-hydrogenase utilizers and *C. jejuni* (R_s_ 0.76). All these correlations were positive.

When correlations were analyzed in the commercial farm chicken group, we observed a strong and positive correlation between methanogens and acetogens (R_s_ 0.64). The other three relationships were weakly correlated: acetogens vs. SRB (R_s_ 0.36), acetogens vs. *L. salivarius* (R_s_ 0.28), and *L. salivarius* vs. *C. jejuni* (R_s_ −0.34). The last pair was the only one negatively correlated.

### 2.5. Sequencing Analysis 

Sequencing analysis of the V3–V4 16S rRNA gene showed that the archaeal population was limited to only two methanogenic genera—*Methanobrevibacter* and *Methanomassiliicoccus*—and their presence was restricted to samples from the free-range and commercial chickens, respectively (Table 1). Analysis of the *Desulfovibrionaceae* family revealed that there were other genera than *Desulfovibrio* in the commercial farm chickens, including *Bilophila* sp. The percentage of *Lachnospiraceae* was the highest of all microbial groups potentially involved in hydrogen sink. Their abundance was the highest in the experimental chickens and lowest in the free-range chickens. A contrasting situation was observed in the *Selenomonadaceae* family, which was limited to only one genus—*Megamonas*. A relatively high abundance of *Coriobacteriia* was also observed, especially in the free-range chickens, and *Bifidobacterium* and *Enterobacterales* in the commercial chickens.

All sequencing data have been deposited in the Sequence Read Archive (SRA) of the National Center for Biotechnology Information (NCBI) repository under BioProject no. PRJNA944200.

## 3. Discussion

The process of hydrogen sink is best described in ruminants. It is known that in the environment of the rumen, the methanogenic archaea dominate over the homoacetogens, especially at low concentrations of H_2_ [2,19]. Only when methanogenesis is suppressed do acetogens take over methanogen’s place [18]. The activity of the remaining hydrogen consumers, i.e., the nitrate- and the sulfate-reducing bacteria—although thermodynamically more favorable than methano- and homoacetogenesis—is usually limited by low nitrate and sulfate concentrations originating from a diet [2]. The impact of dietary shifts on the exact mechanisms responsible for hydrogen metabolism remains largely unknown, and so far has only been studied in the rumen, where methanogenesis is naturally the predominant hydrogen sink [20]. For this reason, any changes observed in ruminants in response to their diet in terms of the methanogen vs. acetogen load, may not translate into chickens, which—as was here demonstrated—rely mainly on acetogenesis.

Acetogenesis was shown to be the preferred way of hydrogen utilization in the hindgut of monogastric herbivores. This phenomenon has been verified in the ceca of rabbits, and the feces of horses [19]. In humans, methanogenesis and homoacetogenesis are the two predominating hydrogen sinks [2]. Studies focusing on hydrogen uptake in chickens are extremely limited [5], and therefore we decided to study this subject.

In this study, the acetogens were generally the most abundant group, followed by methanogenic archaea and [NiFe]-hydrogenase utilizers. The acetogens were more abundant in the cecal contents of both experimental and commercial-farm chickens than they were in the excreta of the free-range chickens. This correlation seems to be justified since *Lachnospiraceae*—probably the largest population of acetogens—has already been shown to be more abundant in the cecal contents than in the excreta in another study [21,22]. The same authors also demonstrated that lactobacilli tend to be more abundant in the excreta than in the ceca, which is also in line with our study. On a side note, the use of excreta in the free-range group of chickens was due to technical and administrative reasons, as cecal samples were simply unavailable.

It is known that the gut microbiota evolves with age after the time of hatching, and, with time, certain taxa outcompete the others [23]. In our study, the only statistically confirmed association between age and the composition of microbiota was observed for *L. salivarius* in the experimental chicken group. Interestingly, lactobacilli were highly abundant in very young chicks, and their prevalence in the 4-day-old chicks was almost as high as the prevalence in the 4-week-old chickens. Moreover, *L. salivarius* was detected in the ceca of three out of five chicks which hatched just hours before the sample collection, which only confirms that colonization of the guts of the chicken embryos by certain bacteria, such as lactobacilli, occurs before hatching, through the egg shell [24]. 

We also observed the tendency (albeit not statistically significant) for a decrease in the level of acetogen counts in relation to the age of the experimental chickens. The acetogens reached their highest counts in week 2 and seemed to be less abundant with time, especially by the end of the 4-week-long experiment. By assuming that *Lachnospiraceae* are the core acetogens, our findings are in line with a study by Videnska et al. [23] that first reported that this family of bacteria accounts for approx. 90% of the chicken gut microbiota at 2 weeks of age, and is being replaced by other bacteria starting with week 3. There were no more statistically significant differences among the age groups, which was most likely caused by the low number of observations, and the fact that many samples tested negative for at least one microbial group.

The methanogenic archaea were not detected in the experimental chickens at any age. This was probably caused by the lack of sources of these microorganisms in the strictly controlled environment with standard, ready-made feed, and communal tap water. Saengkerdsub et al. [25] reported that colonization of chickens with *Methanobacteriales* starts on day 3; however, they used sawdust from the bedding of older chickens with a mature gut microbiota presumably colonized by archaea. The same authors also established the quantity of *Methanobrevibacter woesei* in 56- to 72-week-old chickens at the level of 5.50 and 7.19 log_10_/gram of wet weight of cecal content. The cecal contents of the commercial farm chickens that we studied had a mean value of 4.24 log_10_/gram of wet weight. These differences may be related both to the age of the chickens and the methodology, as we used the *mcrA* gene as a target in real-time PCR, whereas the authors mentioned above used the 16S rRNA gene and cultivation.

*Methanobrevibacter woesei* has been the primary species of methanogens in the chicken gut since approx. 2007, but recently two more species have been discovered—*Methanocorpusculum* and UBA71, both renamed *Candidatus Methanospyradousia* [26]. The sequencing analysis of our samples, i.e., N3, SA2, and SB5, revealed that chickens are also colonized by *Candidatus Methanomassiliicoccus*. Until now, archaea belonging to the genus *Methanomassiliicoccus* have only been found in the gastrointestinal tracts of humans and pigs [27].

It was previously proven that in the presence of excess sulfate, the SRB displace the methanogens [28]. In this study, this did not occur, as SRB were generally rarely detected—mostly in the excreta of free-range chickens—and no correlation between the two groups was ever observed in any experimental configuration. However, the sequencing analysis of our samples revealed certain amounts of bacteria belonging to the genus *Bilophila*. Interestingly, it outnumbered the most common *Desulfovibrio* species [29]. In 2022, the new *Candidatus Bilophila faecipullorum* was reported in the feces of young chickens [26]. Unfortunately, our data did not allow for a full name description of detected sequences.

Until now, as many as 26 distinct hydrogenase subgroups have been discovered, including the hydrogenases that either catalyze the production or the consumption of hydrogen [17]. The bidirectional groups of hydrogenases are also quite common, which makes any research on hydrogen sink even more confusing and difficult to follow. Some research studies report a high abundance of the uptake hydrogenases from *Megamonas* (Selenomonadales); others also report *Wolinella*, *Helicobacter,* and *Campylobacter* to be their source in chickens [5,17,21]. Therefore, this was the starting point for designing an assay targeting the [NiFe]-hydrogenases of these genera as a representation of the H_2_-utilizing hydrogenases. We were able to quantify the uptake [NiFe]-hydrogenases as the third (after acetogens and methanogens) force responsible for hydrogen sink. Studies on ruminants consuming a fiber-rich diet revealed that the amount of detected *hydB* gene was approx. 1.8 times higher in these cows than in those ingesting a starch-rich diet [20]. In our study, this correlation was not observed, as experimental chickens receiving feed consisting of fiber-rich sunflower meal demonstrated a lower abundance of *hydB* compared to the free-range chickens receiving corn as the feed’s major ingredient. As for the sequencing analysis, no *Helicobacter* nor *Wolinella* was ever detected in any of the studied chicken groups. This is especially interesting since there are reports suggesting that *Helicobacter* is often found in commercial broilers [21]. *Helicobacter pullorum* is considered a pathogen, and so is *Campylobacter* spp. [15]. In the present study, *L. salivarius* was shown to negatively correlate with *C. jejuni* and the other [NiFe]-hydrogenase utilizers. The competitive exclusion of lactobacilli and *C. jejuni* in the gastrointestinal tracts has already been well established [1]; however, surprisingly the negative correlation indices reported in this study for the cecal samples were only weak to moderate.

There were also other microbes potentially involved in hydrogen sink detected by 16S rRNA gene sequencing, such as *Eubacterium*, *Enterobacterales*, and *Coriobacteriia*; however, it is difficult to say how many of those were actually hydrogen consumers. In order to fully determine the proportion of the H_2_ uptake genes, further studies should be conducted, e.g., whole-metagenome sequencing would be of great importance.

One last remark of this study relates to the experimental group of chickens. Generally, animals kept in isolated conditions are the key element of many studies [16,23]. However, our results clearly indicated that these chickens were characterized by small diversity of microbionts involved in hydrogen sink, and—as an animal model—were found not optimal for studying any microbial interactions.

## 4. Materials and Methods

### 4.1. Animals

A total of 154 chickens (*Gallus gallus domesticus*) representing three different rearing methods were included in this study: 54 experimental chickens, 50 commercial farm chickens, and 50 free-range chickens. The selected rearing conditions, including the diet, the usage of antibiotics, and the type of environment are listed in Table 2.

#### 4.1.1. The Experimental Chickens

A total of 54 white leghorn chicks were hatched from the SPF (specific pathogen-free) eggs (VALO BioMedia GmbH, Osterholz-Scharmbeck, Germany) in a sanitized incubator (Heka Incubator, Przewoz, Poland) and transferred to sanitized cages where they were kept in standard (non-SPF) conditions. The chickens received water and feed ad libitum. Cages were cleaned daily. The birds did not receive any vaccinations. The chickens were sacrificed on days 0, 4, 7, 14, 21, and 28, either by cervical dislocation (days 0-4) or lethal injection with pentobarbital in a dose of 150 mg/kg (older chickens). The ceca were isolated aseptically and subjected to DNA isolation on the same day. For the metagenome sequencing analysis purposes, chickens were pooled in groups of no more than six individual DNA samples per one pooled sample, and therefore two name entries correspond to one sacrificed chicken group (Table 3).

#### 4.1.2. The Commercial Farm Broiler Chickens

The carcasses of chickens were submitted as soon as possible after killing from the commercial farms located in the Mazovia Province of Poland to the Department of Pathology and Veterinary Diagnostics, Institute of Veterinary Medicine, Warsaw University of Life Sciences (Warsaw, Poland). The chickens were of different ages (Table 4). During necropsy, the ceca of healthy chickens were aseptically removed. No gross lesions were found in the gastrointestinal tract or in other organs.

#### 4.1.3. Rural Free-Range Chickens

In the free-range chickens, the collection of ceca was technically not possible, and therefore samples of excreta had to be included in the study. Samples of fresh excreta were collected from the floor of three henhouses hosting the free-range laying hens. The flocks were located in three different rural areas across Poland (Table 5). All birds had access to large outdoor runs during daytime. If the samples were from the same location, the collections were performed months apart from each other, with new chickens introduced into a flock.

### 4.2. DNA Isolation

Ceca from the experimental and the commercial chickens were longitudinally sectioned to collect 200 mg of the cecal content together with the cecal mucosa, which was scraped off the intestinal wall with the use of sterile scalpel blades. As for the free-range chickens, a total of 200 mg of dropping samples was collected. Then, a DNA isolation procedure was performed according to the protocol described previously [30].

### 4.3. Quantitative Real-Time PCR

The following key functional genes were chosen as targets for the quantitative real-time PCR: the *mcrA* gene encoding methyl-coenzyme M reductase alpha subunit for methanogenic archaea, the *aprA* gene encoding adenosine 5′-phosphosulfate reductase alpha subunit for the SRB, the *acsB* gene encoding acetyl-CoA synthase beta subunit for acetogens, the *hyaB*/*hydB* gene encoding [NiFe]-hydrogenase large subunit for *Wolinella*, *Helicobacter,* and *Campylobacter*, and the *mapA* gene encoding membrane-associated protein for *C. jejuni* alone. The last target microorganism—*L. salivarius*—was detected with the use of the 16S rRNA gene (Table 6). With the exception of the latter, all target genes occur in a single copy per genome. For *L. salivarius*, the results of the real-time PCR were divided by seven (i.e., the number of operons per genome in *L. salivarius*; [31]) to achieve the number of cells per gram of cecal/excreta content. Primers used for *hyaB* were designed de novo for the purpose of this study.

Standard curves were generated by using decimal dilutions, from approx. 10^0^ to 10^6^ copies per reaction of genomic reference DNA. The following reference DNAs were used: *mcrA*+ positive plasmid containing an insert of the *mcrA* sequence fragment from GenBank acc. KF214818.1:976-1447, *Desulfovibrio piger* DSM 749 (SRB), *Ruminococcus gauvreauii* DSM 19829 (acetogen), *Helicobacter cinaedi* DSM 5359 ([NiFe]-hydrogenase carrier), *C. jejuni* 405 (courtesy of Dr. Agnieszka Sałamaszyńska-Guz), and *L. salivarius* 3D (courtesy of Dr. hab. Magdalena Kizerwetter-Świda). The new primer pair targeting the [NiFe]-hydrogenase large subunit was designed in silico by comparing the *hydB*/*hyaB* sequences from the *Wolinella*, *Helicobacter*, and *Campylobacter* group against homologic sequences from other bacteria.

The reaction mixture included 10 μL of RT HS-PCR Mix SYBR A (A&A Biotechnology, Gdynia, Poland), 0.5 μM primers (Table 6), 1 μL of cecal or 0.5 µL of excreta DNA, and water to reach a final volume of 20 μL. Samples were quantified individually, in triplicate. The thermal conditions were first set experimentally in order to achieve the optimal amplification efficiency by using reference DNAs and a gradient PCR. The reaction conditions for each quantitative assay are presented in Table 7. In each reaction, the amplification comprised 45 cycles. The real-time PCR results were calculated into the number of cells in 1 g of the cecal content or excreta.

### 4.4. Statistical Analysis

The Shapiro–Wilk test was used in order to check whether the quantification results of the real-time PCR have a normal distribution. The homogeneity of variance was checked with Levene’s test. Then, a non-parametric Kruskal–Wallis H test was applied to evaluate the statistical significance of variation among the hydrogen consumers and *L. salivarius* regarding the sample type, source, and age of the chickens. Spearman’s rank correlation test was used to measure the strength and direction of the microbial associations grouped by the rearing methods. According to the guidelines for interpretation of Spearman’s rho rank correlation by Prion and Haerling, 2014 [37], the correlations were considered very strong when the values of R_s_ were between 0.81 to 1, strong—0.61 to 0.80, moderate—0.41 to 0.60, weak—0.21 to 0.40, and negligible—0 to 0.20. All statistical analyses were performed in TIBCO Statistica 13.3 (TIBCO Software Inc., Palo Alto, CA, USA) and Microsoft Office Excel 2016 (Redmond, WA, USA).

### 4.5. Sequencing Analysis

The metagenome analysis of archaea and bacteria was performed based on the hypervariable V3–V4 region of the 16S rRNA gene. Only samples with high-quality DNA were selected for sequencing. Samples from the commercial and free-range chickens were sequenced individually, whereas in the case of the experimental chickens, a total of max. six samples from each group were pooled and in this form subjected to sequencing. 

The analysis was outsourced to Genomed S.A. (Warsaw, Poland). In short, the 341F and 785R primers were used together with a Q5 Hot Start High-Fidelity 2X Master Mix (New England Biolabs Inc., Ipswich, MA, USA). The sequencing was performed in the paired-end technology (PE), 2 × 300 nt with Illumina v3 kit by a MiSeq instrument (San Diego, CA, USA), which also performed an initial automatic analysis comprising of demultiplexing and generation of fastq files. The species-specific classification of the reads was performed with the use of QIIME 2 according to the Silva 138 reference sequence database. The following tools were then used: FIGARO for read quality control, Cutadapt for initial data processing, and DADA2 for the selection of ASV (amplicon sequence variant) and further steps of the analysis.

## 5. Conclusions

This work presents the possible routes of hydrogen disposal, pointing out the strong position of acetogenesis as the leading metabolic pathway for hydrogen sink. In this study, we have demonstrated that acetogens were dominant over the other tested groups of hydrogen consumers, whereas the numbers of methanogenic archaea, SRB, and the [NiFe]-hydrogenase utilizers depended on the sample type and rearing conditions. In order to fully determine the role of specific gut microbionts in hydrogen sink, further studies should be conducted.

## Figures and Tables

**Figure 1 ijms-24-06674-f001:**
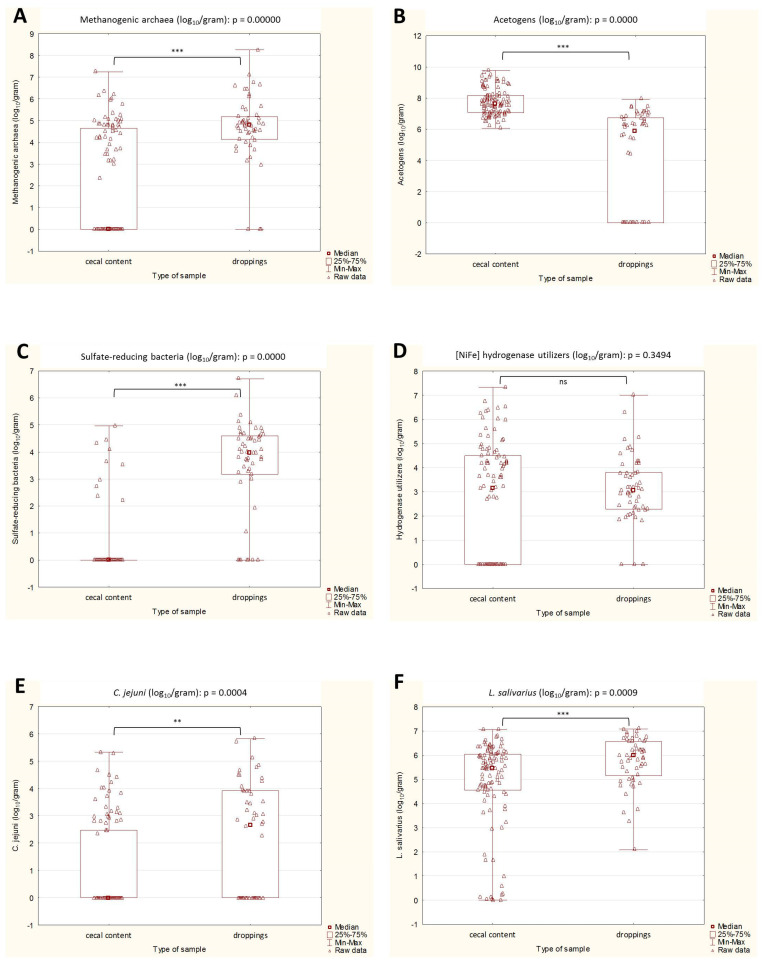
(**A**–**F**). Box plots representing the abundance of selected microbial groups in ceca and excreta samples. Statistical significance between cecal contents and excreta is marked by asterisks. Values of ** *p* < 0.01, and *** *p* < 0.001 were regarded as significant; ns: non-significant.

**Figure 2 ijms-24-06674-f002:**
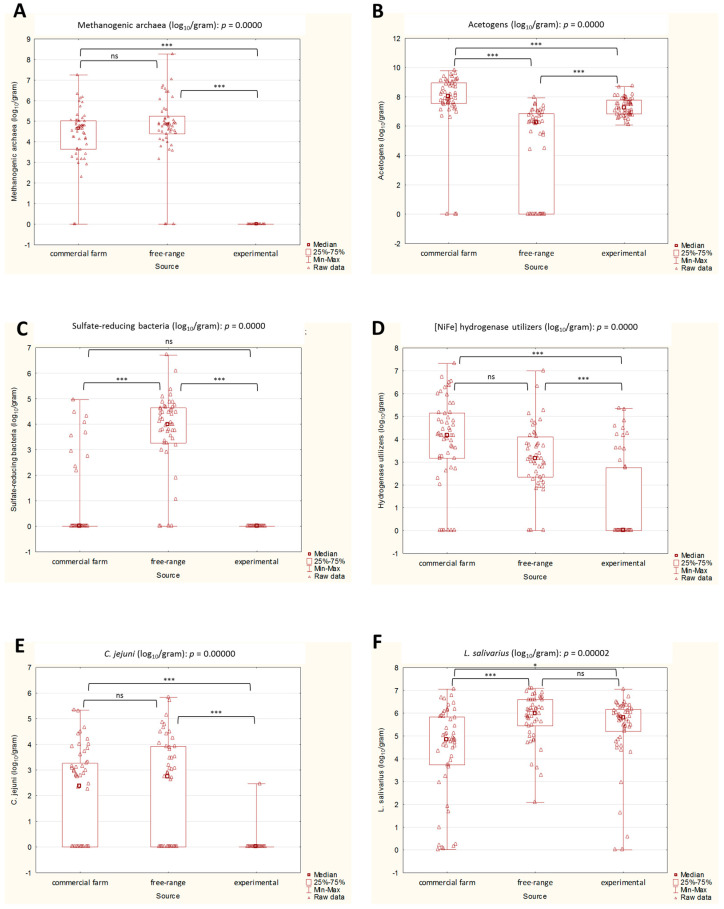
(**A**–**F**). Box plots representing the abundance of selected microbial groups in three rearing systems. Statistical significance among the three groups is marked by asterisks. Values of * *p* < 0.05, and *** *p* < 0.001 were regarded as significant; ns: non-significant.

**Figure 3 ijms-24-06674-f003:**
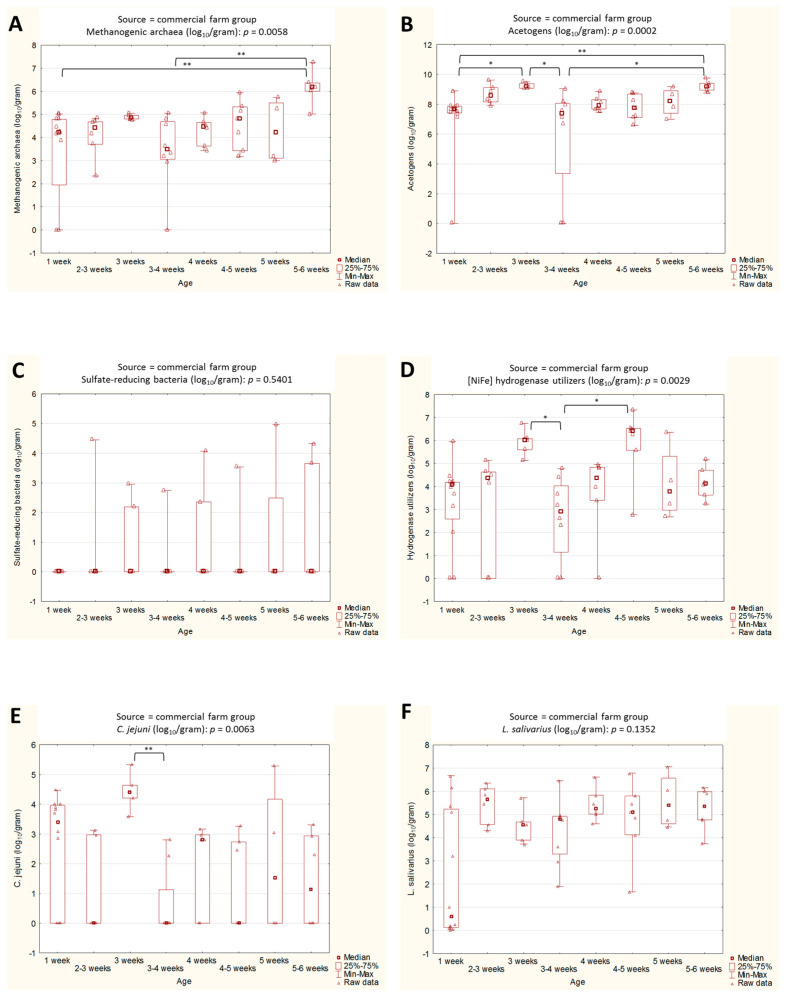
(**A**–**F**). Box plots representing the abundance of selected microbial groups across age groups in the commercial chickens. Statistical significance among the age groups is marked by asterisks. Values of * *p* < 0.05, and ** *p* < 0.01 were regarded as significant. The non-significant results remained unmarked.

**Figure 4 ijms-24-06674-f004:**
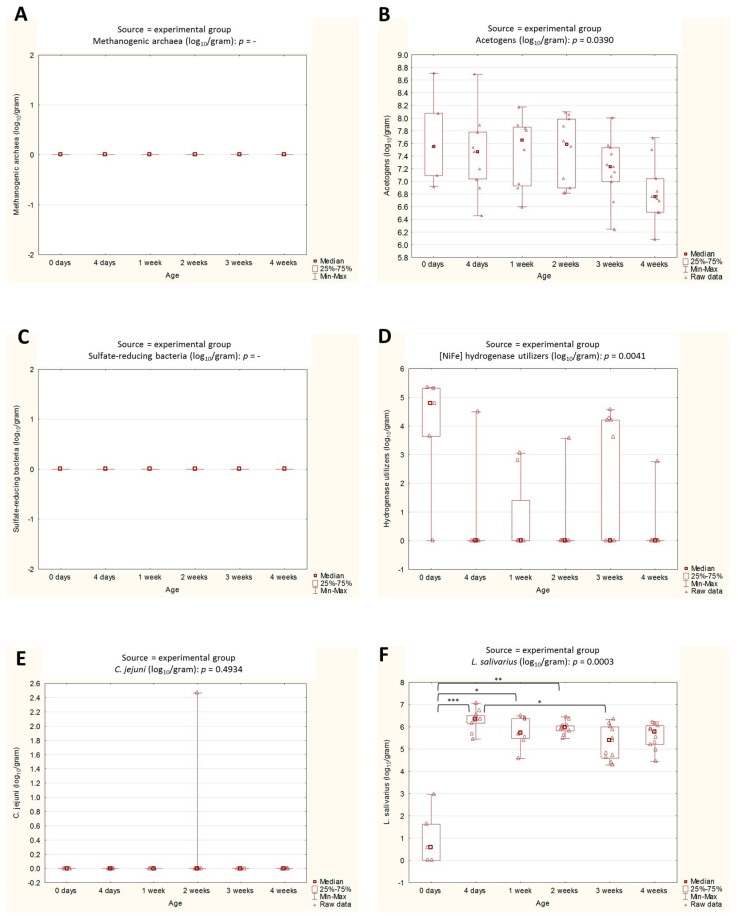
(**A**–**F**). Box plots representing the abundance of selected microbial groups across age groups in the experimental farm chickens. Statistical significance among age groups is marked by asterisks. Values of * *p* < 0.05, ** *p* < 0.01, and *** *p* < 0.001 were regarded as significant. The non-significant results remained unmarked.

**Table 1 ijms-24-06674-t001:** The abundance of selected hydrogen consumers in the total microbiota of samples grouped according to the chicken rearing method.

Microorganism	Catalytic Subunits in Hydrogen Sink [6,17,18]	Experimental Chickens			Commercial Farm Chickens			Free-Range Chickens		
		No. of Hits *	Percentage of Microbiota		No. of Hits *	Percentage of Microbiota		No. of Hits *	Percentage of Microbiota	
			Min. [%]	Max. [%]		Min. [%]	Max. [%]		Min. [%]	Max. [%]
*Methanobrevibacter*	McrA, group 4h and 4i [NiFe]-hydrogenase, AcsB, and FrdA	0/11	0	0	0/30	0	0	4/15	0	1.038
*Methanomassiliicoccus*		0/11	0	0	3/30	0	0.032	0/15	0	0
*Desulfovibrionaceae*	group 1b and 1d [NiFe]-hydrogenase, AprA, DsrA, FrdA, and CydA	0/11	0	0	14/30	0	1.706	3/15	0	0.012
*Desulfovibrio*		0/11	0	0	0/30	0	0	3/15	0	0.012
*Lachnospirales*	AcsB and HydB	11/11	2.56	74.39	30/30	1.799	53.96	12/15	0	11.34
*Peptostreptococcus*	AsrA, AcsB, and CydA	0/11	0	0	1/330	0	0.017	0/15	0	0
*Clostridium*	HydB, AprA, and AsrA	9/11	0	7.228	12/30	0	2.325	9/15	0	0.918
*Eubacterium*	AcsB, HydB, and AsrA	0/11	0	0	12/30	0	1.277	7/15	0	1.245
*Selenomonadaceae*	group 1d [NiFe]-hydrogenase, HydB, FrdA, AprA, NarG, NrfA, DmsA/Tor, and CydA	4/11	0	0.052	4/30	0	20.63	9/15	0	27.98
*Megamonas*	HydB and CydA	0/11	0	0	4/30	0	20.63	9/15	0	27.98
*Coriobacteriia*	group 1i [NiFe]-hydrogenase and DmsA/TorA	4/11	0	1.157	27/30	0	18.59	11/15	0	32.84
*Actinomycetales*	FrdA, NarG, DmsA/TorA, and CydA	0/11	0	0	1/330	0	0.006	10/15	0	0.059
*Corynebacterium*	group 1f [NiFe]-hydrogenase, FrdA, NarG, and CydA	2/11	0	0.113	4/30	0	0.714	7/15	0	7.663
*Bifidobacterium*	FrdA and HydB	1/11	0	0.365	20/30	0	49.79	11/15	0	19.12
*Prevotella*	FrdA and NrfA	0/11	0	0	0/30	0	0	1/15	0	0.041
*Bacteroides*	HydB, FrdA, and NrfA	0/11	0	0	16/30	0	11.1	4/15	0	0.064
*Enterobacterales*	group 1c and 1d [NiFe]-hydrogenase, NarG, NapA, NrfA, DmsA/TorA, and CydA	1/11	0	0.006	30/30	0.006	54.19	9/15	0	22.19
*Pseudochrobactrum*	CydA	0/11	0	0	1/30	0	0	0/15	0	0.156
*Synergistes*	HydB	0/11	0	0	0/330	0	0	2/15	0	1.788

* number of positive samples/number of samples tested.

**Table 2 ijms-24-06674-t002:** The selected rearing parameters.

Chicken Group	Diet	Antibiotics	Rearing Environment
Experimental	Commercial feed for chickens from 1 to 6 weeks old (DKM1; Farmer Sp. z o.o., Biskupice Oloboczne, Poland), composed of sunflower meal, wheat bran, barley, wheat, and corn	No	No contact with the natural environment, other animals, and people.
Free-range	Commercial feed (e.g., Kokoszka-Nioska; ELPOL, Osina Mala, Poland) comprising corn, wheat, barley, black sunflower seeds, gold millet, oat, red millet, yellow peas, green peas, linseed, safflower seeds, and rape seeds; kitchen waste including potatoes, carrots, and eggshells, and worms found in the paddock	No use of antibiotics in the flocks from which the samples were collected. However, the use of antibiotics in the reproductive farms from which the chicks derived cannot be ruled out.	Yes, free-ranging, having contact with other farm and wild animals.
Commercial	A variety of standard commercial feeds varying between farms, adapted to the type and age of chickens (intensive broiler production system).	No growth-promoting antibiotics. Chickens from groups G, H, K, M, N, and SA÷ SD were reared without access to any antibiotics. Chickens from group T received colistin and amoxicillin. No data available for the remaining groups.	Indoor broiler chicken farms with implemented biosecurity procedures.

**Table 3 ijms-24-06674-t003:** Groups of the experimental chickens.

Group Name	Age	Sample Type	No. of Chickens
X0	0	Cecal contents	5
XA1	4 days	Cecal contents	4
XB1	4 days	Cecal contents	5
XA2	1 week	Cecal contents	4
XB2	1 week	Cecal contents	4
XA3	2 weeks	Cecal contents	5
XB3	2 weeks	Cecal contents	5
XA4	3 weeks	Cecal contents	5
XB4	3 weeks	Cecal contents	6
XA5	4 weeks	Cecal contents	6
XB5	4 weeks	Cecal contents	5

**Table 4 ijms-24-06674-t004:** Groups of commercial chickens.

Group Name	Age	Coop Location	Sample Type	No. of Chickens
F	1 week	Farm I	Cecal contents	7
G	3–4 weeks	Farm II	Cecal contents	4
H	1 week	Farm III	Cecal contents (4 samples) and excreta (1 sample)	4
I	4–5 weeks	Farm IV	Cecal contents	6
K	3–4 weeks	Farm III	Cecal contents (1 sample) and excreta (1 sample)	1
M	2–3 weeks	Farm V	Cecal contents	6
N	4 weeks	Farm V	Cecal contents	3
OA	4 weeks	Farm VI	Cecal contents	3
OB	5 weeks	Farm VI	Cecal contents	2
T	3 weeks	Farm IV	Cecal contents	5
SA	5–6 weeks	Farm VII	Cecal contents	3
SB	5–6 weeks	Farm VII	Cecal contents	3
SC	5 weeks	Farm VIII	Cecal contents	2
SD	4–5 weeks	Farm III	Cecal contents	1

**Table 5 ijms-24-06674-t005:** Groups of free-range chickens.

Group Name	Age	Coop Location	Sample Type	No. of Chickens
A	n/a	Henhouse I	Excreta	1
B	n/a	Henhouse I	Excreta	1
E	n/a	Henhouse I	Excreta	2
L	n/a	Henhouse I	Excreta	11
J	n/a	Henhouse II	Excreta	7
P	n/a	Henhouse IIII	Excreta	7
R	n/a	Henhouse I	Excreta	21

n/a—not available.

**Table 6 ijms-24-06674-t006:** Primers used in this study.

Microorganism	Target Gene	Forward Primer 5′–3′ Sequence	Reverse Primer 5′–3′ Sequence	Amplicon Length [bp]	Reference
Methanogenic archaea	*mcrA*	CTTGAA RMTCAC TTCGGT GGWTC	CGTTCA TBGCGT AGTTVG GRTAGT	Approx. 270	[32]
SRB	*aprA*	TGGCAG ATCATG ATYAAY GG	GGCCGT AACCGT CCTTGA A	Approx. 385	Forward primer—modified [33]; reverse primer—modified [34]
Acetogens	*acsB*	CTBTGY GGDGCI GTIWSM TGG *	AARCAW CCRCAD GADGTC ATIGG *	216	[18]
selected [NiFe]-hydrogenase utilizing bacteria	*hyaB*/*hydB*	ATTGAA GTTGTT GTTGAT GAWAAY AATGT	AGMCCA ATCAAG CCCRTG	300	This study
*C. jejuni*	*mapA*	CTATTT TATTTT TGAGTG CTTGTG	GCTTTA TTTGCC ATTTGT TTTATT A	589	[35]
*L. salivarius*	16S rDNA	TACACC GAATGC TTGCAT TCA	AGGATC ATGCGA TCCTTA GAGA	138	[36]

* I—inosine.

**Table 7 ijms-24-06674-t007:** Temperature settings and DNA standards used for the absolute quantification of each group of microorganisms.

Real-Time PCR Step	Methanogenic Archaea	SRB	Acetogens	[Nife]-Hydrogenase Utilizing Bacteria	*C. jejuni*	*L. salivarius*
Initial Denaturation	95 °C—5 min					
Denaturation	94 °C—20 s	94 °C—20 s	94 °C—20 s	94 °C—20 s	94 °C—20 s	94 °C—20 s
Annealing	60 °C—20 s	62 °C down to 60 °C after first 10 cycles with 0.1 °C/s decreasing rate—20 s (touchdown PCR)	61 °C—20 s	64 °C—20 s	58 °C—20 s	68 °C—20 s
Elongation	72 °C—20 s	72 °C—30 s	72 °C—20 s	72 °C—20 s	72 °C—20 s	72 °C—20 s
Signal acquisition *	81 °C—20 s + Acq	89°C—20 s + Acq	80 °C—20 s + Acq	81 °C—20 s + Acq	79 °C—20s + Acq	82 °C—20 s + Acq
Melt analysis *	95 °C—5 s, then 60 °C—1 min, and 95 °C—continuous Acq with ramp rate 0.11 °C/s					

* Acq—acquisition of fluorescence signal.

## Data Availability

Data are contained within this article and Appendix A. The sequencing data have been deposited in the Sequence Read Archive (SRA) of the National Center for Biotechnology Information (NCBI) repository under BioProject no. PRJNA944200.

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
