# Peer review of "Microorganisms Involved in Hydrogen Sink in the Gastrointestinal Tract of Chickens"

_ijms, 2023, doi:10.3390/ijms24076674_

Round 1

Reviewer 1 Report

Congratulations to the authors, the matter is very interesting, especially for the people who think livestock is a problem over methane global emission but isn’t true. In my opinion, methane emission is correlated with the type of diet, this is information very important and is missing in the paper. The paper brought innovative information, of course.

The paper can be accepted, but I think the authors can explore your data if the includes the diets. However, I’m worried about some comparisons between commercial broilers and free range, In this case, chickens in free range look better than commercial broilers…but I don’t believe this especially if the chickens have access to grass.

The authors need to review the English, please. I’m putting some errors below… review all the text.

For example, in Line: 32: The breakdown

Line 38: remove the word “the” livestock.

Line 41: remove the word “the” climate.

The introduction is so big… having some information isn’t necessary. In my opinion, a good introduction needs to be in a maximum of 4 paragraphs.

Line 42-45: It was estimated that approx. 5% of the methane global emission 42 comes from the animal production branch of the industry [3].

5% is very few, I don’t believe the methane global emission is a problem for livestock.

Line 43-45.  Remove this sentence.

Line 79: What is SRB? You need to describe, not only the abbreviation.

Line 84-86: The necessity of elimination of C. jejuni from chicken rearing is a matter of public 84 health because campylobacteriosis is the most common cause of foodborne 85 gastrointestinal infections in humans and has been so since 2007.

Who said that? Put reference.    

Line 101: Remove the word current, this study is current now…but if someone read ten years from now…isn’t current.

Lien 103: Where is the point at the end of the paragraph?

Line 104-113: Why these two paragraphs are in the introduction session? This is part of the mythology. Remove.

I prefer the paper to follow the formation: introduction, material and methods, results, and discussion.

317. remove the word broiler.

322-323: They had unrestricted access to standard chicken feed and water.

Replay by: The chickens received water and feed ad libitum.

323: The animals.

Replay by the birds

323:Didn’tt

324-325: Ok, how many chickens have you killed by day?  Why you killed only five chickens on day 0?

331: Remove the word broiler

Line 335: Table 3

In Table 2 authors used days and in table 3 weeks….please patronize this to facilitate the knowledge of the reader.

Line 398: Isn’t about the results, is about correlation.

correlations were considered very strong when the values of Rs were between 0.81 to 1, strong – 0.61 to 0.80, moderate 399 – 0.41 to 0.60, weak – 0.21 to 0.40, and negligible – 0 to 0.20.

Line 116: Excreta not dropping! Replay this word all the paper.

Como os autores fazem essa comparação? Porque a dieta oferta para as aves influencia no tipo de microbiota e consequentemente em seus produtos.

Line 191-194: Isn’t a Strong correlation...strong correlation is between 0.81 til 1.0.

Reviewers this value conforms you wrote in the statistical analysis.

A discussão é baseada no tipo de dieta e os autores não avaliaram as dietas? Acho que muitas das diferenças encontradas entre sistemas está relacionada com a dieta e no caso aves do grupo experimental com a dieta e a limpeza que era realizada todo o dia o que eu subentendo que os animais foram criados em gaiolas.

Line 232-234: This correlation seems to be justified since  Lachnospiraceae – probably the largest population of acetogens – has already been shown to be more abundant in cecal contents than in droppings in another study [4,33].

Why?

Line 234-236: The same authors also demonstrated that lactobacilli tend to be more abundant in droppings than in ceca, which is also in line with our study.

Why?

Line 237-345: Why this occurred? Usually, young animals have more than lactobacilli because they there is immature intestinal.

Line 347: Removed. This is a result, not a discussion.

Line 161: Remove The study of….

Saengkerdsub et al. [35] reported that colonization of chickens …..

Line 267: These differences may be 267 related to the age of the chickens. How can you say this, if you don’t know the age of the chickens in free-range??????

Line 278: Didn’t

Line 282: Only a year ago a study. Remove this!

In 2022 was reported the new Candidatus Bilophila faecipullorum was in young chicken feces [36].

Line 196-276: Helicobacter pullorum is considered a pathogen, but so is Campylobacter spp. [18].

What do you want to mean by this phrase?  When we use the but means something contradictory or an exception. Don’t make sense to me that this phrase.

Reference

Review the reference, the format between the references is not the same

Reviewer 2 Report

In this study, the authors were to assess the quantity and quality of microorganisms involved in the hydrogen uptake in chickens with the use of real-time PCR and metagenome sequencing under different sample types, different rearing methods, and different age conditions. This study was well conducted and written. However, there are a few issues that need to be addressed.

This experiment is just to assess the quantity and quality of microorganisms involved in the hydrogen uptake in chickens without being associated with the host. However, this article has been submitted in the Special Issue "Gut Microbiota–Host Interactions: From Symbiosis to Dysbiosis 2.0". I'm not sure whether this article is suitable for this special issue, which will be decided by the editorial department.

The author did not explain the significance of the research in the article in the Abstract and Conclusion sections.

As the author introduced in the Introduction section hydrogen sink are important because of the effectiveness of livestock production and climate change. Is hydrogen sink related to the gut health or host health of chickens? If the results of the hydrogen sink-related index could contact health problems in chicken production?

Sequencing data needs to be uploaded on an open access data repository (e.g. NCBI SRA) and the accession number has to be added here.

Line 209. Change “Bifidobacterium” to “Bifidobacterium”.

Table 5. Is the primer from 5’ to 3’?

Author Response

Dear Reviewer,

thank you for your work and valid remarks. Changes have been made accordingly.

Remark 1: This experiment is just to assess the quantity and quality of microorganisms involved in the hydrogen uptake in chickens without being associated with the host. However, this article has been submitted in the Special Issue "Gut Microbiota–Host Interactions: From Symbiosis to Dysbiosis 2.0". I'm not sure whether this article is suitable for this special issue, which will be decided by the editorial department.

Response: In this study we tried to demonstrate which of the group of microbes that naturally colonize the chicken intestines may be responsible for the hydrogen uptake. From the host’s point of view, acetogens seem to be the most preferred group of the hydrogen utilizers, as so far – unlike the remaining groups – they have never been connected to any gastrointestinal disorders. Perhaps we did not put enough effort to demonstrate the microbiota-host interactions in our study, therefore we have made some improvements to the manuscript, hoping that it will now fit better the theme of this special issue.

Remark 2: The author did not explain the significance of the research in the article in the Abstract and Conclusion sections.

Response: It has been explained more precisely in both of these sections.

Remark 3: As the author introduced in the Introduction section hydrogen sink are important because of the effectiveness of livestock production and climate change. Is hydrogen sink related to the gut health or host health of chickens? If the results of the hydrogen sink-related index could contact health problems in chicken production?

Response: Hydrogen sink is indirectly related to the maintenance of the host’s health, as it regulates the speed and intensity of the gut fermentation. Production of the short chain fatty acids (SCFAs) is influenced by the hydrogen partial pressure among others. It is known that SCFAs, mostly butyrate, are readily used by the host’s intestinal epithelium. Therefore, we believe that without the hydrogen sink, the condition of intestines would surely worsen. Moreover, the hydrogen sink helps to maintain the proper structure of the gut microbiome, and gut homeostasis, as high concentrations of H2 affect both the hydrogen-producing and the non-producing bacteria. There is also a direct link between the dysbiosis involving methanogenic archaea and overall microbial dysbiosis affecting the host’s health.

In a broader sense, the hydrogen buildup, would result in microbial imbalance, and would deprive its hosts of useful nutrients, causing drop in the animal production indices. Hydrogen sink is supposed to prevent it. Perhaps we might not have exhausted this subject in the Introduction section, which we have now fixed. This section has been redesigned and extended accordingly.

Remark 4: Sequencing data needs to be uploaded on an open access data repository (e.g. NCBI SRA) and the accession number has to be added here.

Response: The sequencing data have been uploaded to the NCBI SRA under BioProject no. PRJNA944200. The BioProject and associated SRA metadata are available at https://dataview.ncbi.nlm.nih.gov/object/PRJNA944200?reviewer=cqab7uqpmgbs62kl8mgo53ovb3

Remark 5: Line 209. Change “Bifidobacterium” to “Bifidobacterium”.

Response: It has been changed.

Remark 6: Table 5. Is the primer from 5’ to 3’?

Response: Yes, this information has been added.

Reviewer 3 Report

Abstracts

Line 12: Please put the” before improvement

Line 14: Please change “microbionts” to “microbiomes”

Line 15: Please change “microbionts” to “microbiomes”

Line 15: In chickens, the microbiomes are responsible….

Line 18-19: Please indicate how many chickens in your study

Line 19-24: Due feed intake affects the gut microbiome population, You should compare the parameters among groups (free range, commercial, and experimental chicken)

This study is very poor. You should provide the feed ingredients data of each groups, due feed is the most essential factor associated with microbiome population in gut. Then, the links among groups could answer the result.

Author Response

Dear Reviewer,

thank you for your work and valid remarks. Changes have been made accordingly.

Remark 1: Abstracts

Response: The abstract has been revised.

Remark 2: Line 12: Please put the” before improvement

Response: This sentence has been removed as a result of abstract revision and redesign.

Remark 3: Line 14: Please change “microbionts” to “microbiomes”

Response: This sentence has been deleted to provide more information about the significance of our findings in place of the less important data.

Remark 4: Line 15: Please change “microbionts” to “microbiomes”

Response: This sentence has been deleted as a result of abstract revision.

Remark 5: Line 15: In chickens, the microbiomes are responsible….

Response: This sentence has also been deleted as a result of abstract revision.

Remark 6: Line 18-19: Please indicate how many chickens in your study

Response: The exact numbers have been provided.

Remark 7: Line 19-24: Due feed intake affects the gut microbiome population, You should compare the parameters among groups (free range, commercial, and experimental chicken)

This study is very poor. You should provide the feed ingredients data of each groups, due feed is the most essential factor associated with microbiome population in gut. Then, the links among groups could answer the result.

Response: Thank you for this remark – diet is important when it comes to the gut microbial composition. For this reason we have described in detail – with all the information we had access to – the feed parameters in one additional table in Materials and Methods, together with some other rearing parameters. Moreover, more attention has been put on the diet in the Discussion section. However we must clarify that we didn’t evaluate the diet but the rearing conditions as a whole. In our research we have tried to reach beyond the diet itself. We studied the entire rearing methods as a combination of diet, the use of antibiotics, agrotechnical measures and the access to the natural environment, from which the chickens may acquire the food and microbes. We tried to understand how the gut microbiota shapes in the real life, in the field, not just in the experimental conditions in response to a selected experimental factor. That is why we confronted chickens kept in strictly controlled experimental conditions (without the access to the outside world) with those that are either reared in a countryside by small farmers as free-range chickens or in large-scale commercial farms, where the main focus is being put on the production indices. Both of these systems represent the poultry manufacturing in Poland.

Round 2

Reviewer 1 Report

Congratulations to the authors, in my opinion, the paper can be accepted after the alterations. 

The research is interesting, but we need to be more careful when we talk about free range, I think we need to do more research on this topic.